# Community control strategies for scabies: A cluster randomised noninferiority trial

Myra Hardy[1,2], Josaia Samuela[3], Mike Kama[3], Meciusela Tuicakau[3], Lucia Romani[4], Margot J. Whitfeld[5], Christopher L. King[6], Gary J. Weil[7], Tibor Schuster[8], Anneke C. Grobler[2,9], Daniel Engelman[1,2], Leanne J. Robinson[10], John M. Kaldor[4], Andrew C. Steer[1,2]*

1 Tropical Diseases, Murdoch Children's Research Institute, Melbourne, Victoria, Australia, 2 Department of Paediatrics, University of Melbourne, Melbourne, Victoria, Australia, 3 Fiji Ministry of Health and Medical Services, Suva, Fiji, 4 Kirby Institute, University of New South Wales, Sydney, New South Wales, Australia, 5 St Vincent's Hospital, University of New South Wales, Sydney, New South Wales, Australia, 6 Center for Global Health and Diseases, Case Western Reserve University and Veterans Affairs Medical Center, Cleveland, Ohio, United States of America, 7 Department of Medicine, Washington University, St. Louis, Missouri, United States of America, 8 Department of Family Medicine, McGill University, Montreal, Quebec, Canada, 9 Clinical Epidemiology and Biostatistics Unit, Murdoch Children's Research Institute, Melbourne, Victoria, Australia, 10 Vector-borne Diseases and Tropical Public Health, Burnet Institute, Melbourne, Victoria, Australia

* andrew.steer@rch.org.au

**Data Availability Statement:** All relevant data are within the manuscript and its Supporting information files.

## Abstract

### Background

Scabies is a neglected tropical disease hyperendemic to many low- and middle-income countries. Scabies can be successfully controlled using mass drug administration (MDA) using 2 doses of ivermectin-based treatment. If effective, a strategy of 1-dose ivermectin-based MDA would have substantial advantages for implementing MDA for scabies at large scale.

### Methods and findings

We did a cluster randomised, noninferiority, open-label, 3-group unblinded study comparing the effectiveness of control strategies on community prevalence of scabies at 12 months. All residents from 35 villages on 2 Fijian islands were eligible to participate. Villages were randomised 1:1:1 to 2-dose ivermectin-based MDA (IVM-2), 1-dose ivermectin-based MDA (IVM-1), or screen and treat with topical permethrin 5% for individuals with scabies and their household contacts (SAT). All groups also received diethylcarbamazine and albendazole for lymphatic filariasis control. For IVM-2 and IVM-1, oral ivermectin was dosed at 200 μg/kg and when contraindicated substituted with permethrin. We designated a noninferiority margin of 5%.

We enrolled 3,812 participants at baseline (July to November 2017) from the 35 villages with median village size of 108 (range 18 to 298). Age and sex of participants were representative of the population with 51.6% male and median age of 25 years (interquartile range 10 to 47). We enrolled 3,898 at 12 months (July to November 2018). At baseline, scabies prevalence was similar in all groups: IVM-2: 11.7% (95% confidence interval (CI) 8.5 to

**Funding:** The study was supported by a grant from the Bill & Melinda Gates Foundation to Washington University (OPPGH5342; G.J.W.). This study also received financial support from the Coalition for Operational Research on Neglected Tropical Diseases (A.C.S), which is funded at The Task Force for Global Health primarily by the Bill & Melinda Gates Foundation (OPP1190754), by UK aid from the British government, and by the United States Agency for International Development through its Neglected Tropical Diseases Program. Ivermectin was purchased at a reduced price from Merck Sharp Dohme (Australia) Pty. Ltd. The funders had no role in study design, data collection and analysis, decision to publish, or preparation of the manuscript.

**Competing interests:** I have read the journal's policy and the authors of this manuscript have the following competing interests: M. H. and G. J. W. report grants from the Bill & Melinda Gates Foundation. M. H. reports a grant from Australian Centre for the Control and Elimination of Neglected Tropical Diseases, National Health Medical Research Council Centre for Research Excellence, during the conduct of the study. All other authors have declared that no competing interests exist.

**Abbreviations:** AIM, Azithromycin Ivermectin MDA; CI, confidence interval; FIT, Fiji Integrated Therapy; ICC, intracluster correlation; IVM-1, one-dose ivermectin-based MDA; IVM-2, two-dose ivermectin-based MDA; MDA, mass drug administration; SAT, screen and treat with 1-dose permethrin to index cases of scabies and their household contacts; SHIFT, Skin Health Intervention Fiji Trial; WHO, World Health Organization.

16.0); IVM-1: 15.2% (95% CI 9.4 to 23.8); SAT: 13.6% (95% CI 7.9 to 22.4). At 12 months, scabies decreased substantially in all groups: IVM-2: 1.3% (95% CI 0.6 to 2.5); IVM-1: 2.7% (95% CI 1.1 to 6.5); SAT: 1.1% (95% CI 0.6 to 2.0). The risk difference in scabies prevalence at 12 months between the IVM-1 and IVM-2 groups was 1.2% (95% CI −0.2 to 2.7, $p = 0.10$). Limitations of the study included the method of scabies diagnosis by nonexperts, a lower baseline prevalence than anticipated, and the addition of diethylcarbamazine and albendazole to scabies treatment.

## Conclusions

All 3 strategies substantially reduced prevalence. One-dose was noninferior to 2-dose ivermectin-based MDA, as was a screen and treat approach, for community control of scabies. Further trials comparing these approaches in varied settings are warranted to inform global scabies control strategies.

## Trial registration

Clinitrials.gov NCT03177993 and ANZCTR N12617000738325.

## Author summary

### Why was this study done?

- Ivermectin-based mass drug administration (MDA) has been successful in reducing community prevalence of scabies in endemic island settings.

- Ivermectin has no ovicidal activity against scabies; therefore, a second dose 7 to 14 days after the first (when eggs have hatched) is recommended for individual treatment and has been adopted for MDA protocols.

- A second dose increases the cost and complexity of MDA and complicates integration of scabies control into 1-dose programmes for other neglected tropical diseases.

- Studies of scabies in populations receiving ivermectin as part of MDA for lymphatic filariasis have suggested that 1 dose of ivermectin may be adequate for community control of scabies.

### What did the researchers do and find?

- We found 1-dose ivermectin-based MDA was noninferior to 2 dose for reducing scabies prevalence at 12 months.

- A screen and treat approach with direct dispensing of permethrin to participants with scabies and their household contacts was also effective in reducing scabies prevalence.

**What do these findings mean?**

- Our findings support the potential for a 1-dose ivermectin-based MDA strategy for scabies control in endemic island settings.

- While a screen and treat approach is also effective in this research context, such a strategy is unlikely to be feasible at scale.

- Replication of our findings in larger populations, in non-island locations, and with varied scabies prevalence is needed before a 1-dose ivermectin-based MDA strategy can be recommended for scabies control.

## Introduction

Scabies is a pruritic, papular rash caused by the mite *Sarcoptes scabiei* var. *hominis*. The infestation is transmitted by human-to-human skin contact and therefore is more common in crowded dwellings, which arise most often in resource-limited areas [1]. Itching and associated scratching due to scabies can lead to considerable morbidity ranging from sleep disturbance through to secondary bacterial infections and their sequelae [1]. For these reasons, and because of emerging evidence of successful control interventions, scabies was included on the World Health Organization's (WHO) list of neglected tropical diseases in 2017 [2]. Multiple studies have documented a high burden of scabies in Pacific island countries, including in Fiji [3–5].

For several key neglected tropical diseases, mass drug administration (MDA) is the primary control strategy in endemic settings. For scabies, the Skin Health Intervention Fiji Trial (SHIFT) showed that 2-dose ivermectin-based MDA (with permethrin treatment for individuals where ivermectin was contraindicated) was a highly effective intervention for scabies and was superior to MDA using only permethrin [5]. After 1 round of 2-dose ivermectin-based MDA, scabies prevalence reduced from 32.1% at baseline to 1.9% at 12 months and was sustained out to 24 months [6]. This finding was replicated on a larger scale in the Solomon islands, in the Azithromycin Ivermectin MDA (AIM) trial, in which ivermectin-based MDA was coadministered with azithromycin for trachoma in a study population of over 26,000 [7,8].

Ivermectin is active against the adult scabies mite but not its eggs [9], and, therefore, 2 doses of ivermectin 7 to 14 days apart are recommended for treatment of individuals with scabies [10]. The WHO Informal Consultation on a Framework for Scabies Control recommends a 2-dose ivermectin-based MDA for community control based on evidence from the SHIFT and AIM trials [5,7,11]. However, the requirement for 2 doses greatly increases programme costs and duration, participation burden for the community, and is difficult to integrate with other neglected tropical disease MDA programmes, which are all 1-dose.

A retrospective study in Zanzibar found a substantial reduction in scabies presentations to clinics following single, annual rounds of ivermectin (with albendazole) for lymphatic filariasis [12]. However, there has been no previous study specifically comparing 1-dose versus 2-dose ivermectin-based MDA for scabies. This research gap was highlighted as a priority in the WHO consultation on scabies control [11].

The Fiji Integrated Therapy (FIT) study was an open-label, cluster randomised trial, implemented as one component of a 5-country trial comparing the safety and efficacy of MDA for lymphatic filariasis using diethylcarbamazine and albendazole, adding ivermectin to the

combination [13,14]. For the Fiji component, we adapted the international study design for filariasis to nest a trial for scabies control.

## Methods

### Study design

This was a 3-group, open-label, cluster randomised, noninferiority trial comparing the effectiveness of 3 community interventions for scabies control (S1 Fig), with the village as unit of randomisation or cluster, nested within a safety and efficacy study for control of lymphatic filariasis, as described previously [13,14]. Participants in the first group (IVM-2) were offered ivermectin-based treatment plus diethylcarbamazine and albendazole followed by a second dose of ivermectin-based treatment 8 days later. Participants in the second group (IVM-1) were offered the same treatment as IVM-2, without the second dose of ivermectin-based treatment. In the third group (screen and treat, or SAT), participants were offered diethylcarbamazine and albendazole and screened for scabies by clinical examination. Those with scabies, and their household contacts, were provided 1 dose of topical permethrin, the current standard of care for case treatment in Fiji [15].

The study protocol (S1 Protocol) was approved by relevant Fijian governmental departments, the Fiji National Health Research and Ethics Review Committee (reference 2016.81. MC), and the Royal Children's Hospital Melbourne Human Research Ethics Committee (reference 36205).

### Participants

The trial was conducted in 2017 and 2018 in 35 villages on Rotuma and Gau, 2 remote islands within the Eastern Division of Fiji (Fig 1). MDA using ivermectin or permethrin has not previously been implemented on these islands. All residents were eligible to participate and approached through their local village administrative structures. Community engagement was undertaken in each village, including an interactive presentation explaining the study, facilitated by village leaders and local health staff. Key points included treatment allocation, individual consent, screening for infections, exclusion criteria for treatment, the need for directly observed treatment for oral medication, and study visit schedules. All residents were invited to the community central meeting place to participate. Written consent was required from all participants aged over 12 years, and written parental/guardian consent for those aged less than 18 years.

A second enrolment of residents took place 12 months after MDA. In order to measure the prevalence of scabies and impetigo in the entire community, residents not present at baseline were eligible.

### Randomisation and masking

All 17 villages on Rotuma and 18 villages on Gau agreed to participate prior to treatment allocation. Randomisation of the 35 villages (clusters) was generated and allocated by an independent statistician using Stata software in a 1:1:1 ratio stratified by island. Stratification by island was implemented by randomising villages on an island equally to the treatment arms and separately for each island. No villages dropped out after allocation. There was no allocation concealment and no blinding of participants or the study team involved in recruitment, clinical examination, treatment assessment, or analysis.

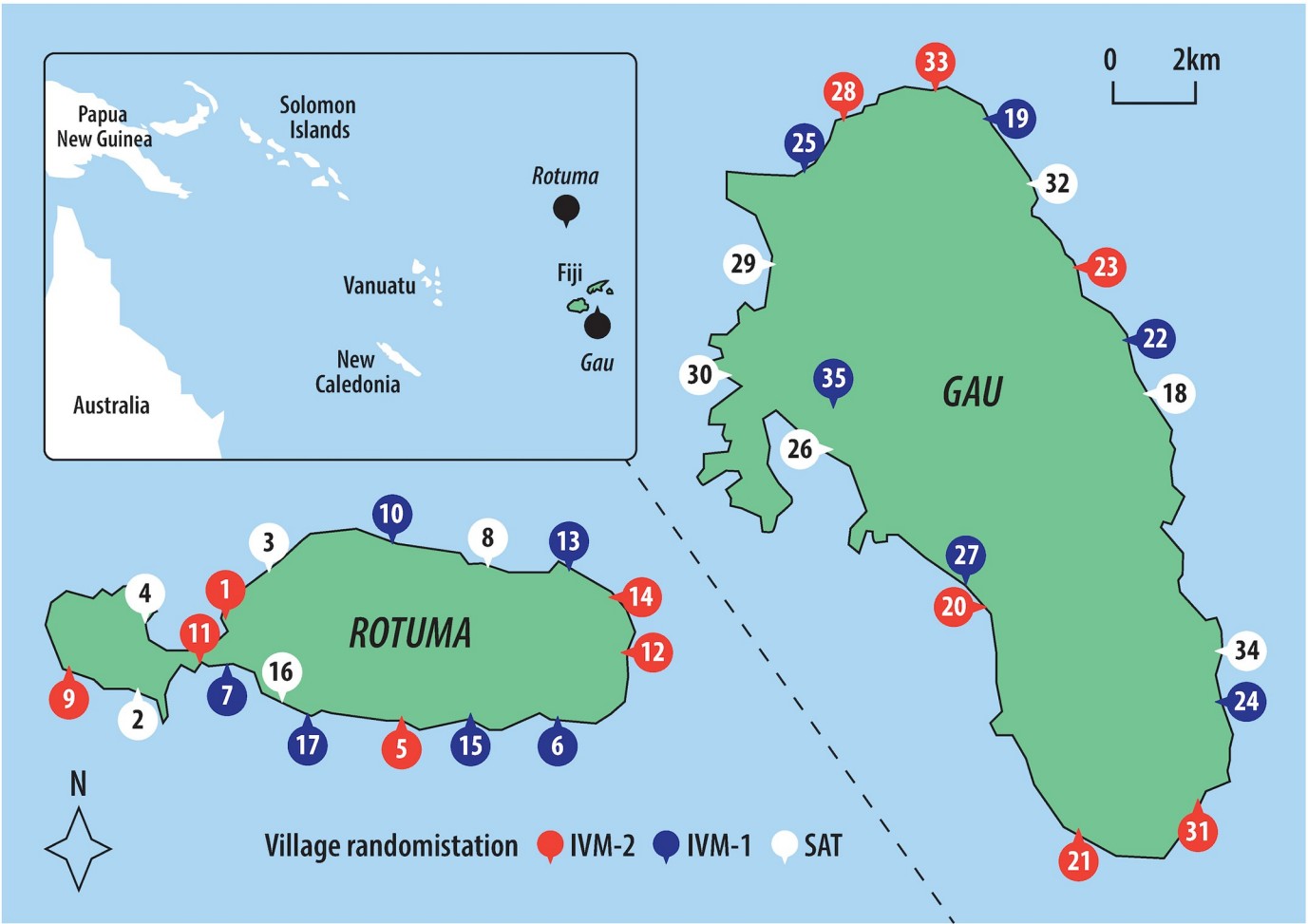

**Fig 1. Map of study sites, village locations, and treatment allocation.** IVM-1, one-dose ivermectin-based MDA; IVM-2, two-dose ivermectin-based MDA; MDA, mass drug administration; SAT, screen and treat with 1-dose permethrin to index cases of scabies and their household contacts. Made with reference to Natural Earth (naturalearthdata.com) and The Pacific Community (SPC) Statistics for Development Division (pacificdata.org/data/dataset/2017_fji_phc_admin_boundaries).

## Procedures

At the baseline and 12-month visits, all participants underwent skin examination by one of 2 trained nurses. Both nurses were recruited from the Dermatology Hospital in Fiji and completed half-day, classroom-based training in the clinical diagnosis of scabies. Supervision was provided by a paediatrician during the initial fieldwork at both time points. All exposed skin areas were assessed. Unexposed areas were examined in a separate, private area if the nurse suspected scabies or at the participant's request. Scabies was diagnosed by identification of typical scabies lesions [5,16]. Impetigo was defined as skin sores that were pus filled, had overlying crusted pus, or had surrounding erythema [16]. Individuals with impetigo were referred to the local health clinic for treatment.

Oral ivermectin was dosed according to weight aiming for 200 μg/kg using whole 3 mg tablets only and administered under direct observation by a study team member (S1 Table). Exclusion criteria for ivermectin treatment were the following: age less than 5 years, weight under 15 kg, pregnancy, breastfeeding within 7 days of delivery, severe illness, or known

allergy to ivermectin. For those excluded from ivermectin, treatment with topical permethrin cream 5% (Glenmark Pharmaceuticals) was substituted as 2 doses in IVM-2 or 1 dose in IVM-1. In the SAT group, 1 dose of permethrin cream was dispensed to participants diagnosed with scabies and each of their household contacts. Participants treated with permethrin in all groups were advised by study staff to apply over the whole body for 8 hours overnight (4 hours for children aged less than 2 months) before washing off, but application was not directly observed. Diethylcarbamazine and albendazole were given to eligible participants in all groups, according to schedules and inclusion criteria described previously (S1 Table) [13,14].

The procedures at the 12-month visit were similar to those at baseline. Changes included surveying participants as to whether they had left the island during the preceding year. All eligible were offered coadministered treatment with 1 dose of ivermectin, diethylcarbamazine, and albendazole, which had by then become the Fijian control strategy for lymphatic filariasis based on updated WHO guidelines [17]. Individuals with scabies and their household contacts were provided with 1 dose of permethrin cream.

## Outcomes and statistical analysis

The primary outcome measure was the absolute reduction in community prevalence of scabies and impetigo between baseline and 12 months. We assumed an estimated scabies prevalence range of 25% to 35% across clusters, based on previous Fijian surveys [4,5], inducing an intracluster correlation (ICC) between 0.03–0.06 adopting the Fleiss-Cuzick estimator for ICC (S1 Methods) [18]. We assumed an absolute reduction of scabies of 29% in IVM-2 group, 22% in IVM-1 group, and 15% in the SAT group, based on findings from SHIFT [5]. Utilising a Monte Carlo simulation tool with 1,000 simulations and Bonferroni-adjusted one-sided confidence intervals (CIs) that maintain a global type I error of 5%, we estimated that 24 clusters of 100 with balanced random allocation of exposure would be sufficient to achieve at least 80% power for comparison of treatment effect between any of the 2 groups.

We used the number of participants examined at baseline and 12 months as the denominators for scabies community prevalence calculations. Participants' data were analysed in the treatment group of their village they were resident in. Overall point prevalence estimates with 95% CI at baseline and 12 months were calculated, taking account of clustering by village using Taylor series linearization of complex sample variance and stratification by island. To calculate the absolute reduction in prevalence between the 2 time points, the 12-month prevalence was subtracted from the baseline prevalence for every village, before calculating the mean of the difference for each treatment group. Since the aim was to compare strategies for community control of scabies, we used the 2-dose ivermectin-based treatment group as the reference group for most of the analyses, with the rationale that this is the current recommendation for scabies MDA. Risk differences of scabies prevalence at 12 months between any 2 groups were calculated using generalised linear models with binomial distribution and identity link that adjusted for clustering by village and stratification by island. We considered a treatment approach to be noninferior if the upper limit of the two-sided 95% CI for risk difference between any 2 groups was 5% or less. We calculated the population attributable risk of impetigo from scabies and used bootstrapping with repetitions of 5,000 to generate 95% CIs.

The broader trial included evaluation of the safety of ivermectin, diethylcarbamazine, and albendazole compared to diethylcarbamazine and albendazole, as well as the impact of these treatments on lymphatic filariasis at 12 and 24 months, which have been reported previously [13,14]. In addition, the effect on soil-transmitted helminths at 12 months and the acceptability of MDA in Fiji will be reported separately.

Data were analysed using Stata software version 14.2. The trial was prospectively registered (Clinitrials.gov NCT03177993 and ANZCTR N12617000738325).

## Results

We enrolled 3,812 participants at baseline between 13 July to 14 November 2017, and 3,898 at 12 months between 24 July to 19 November 2018 (Fig 2), covering 82% of the recorded resident population at both visits, similar between groups (Table 1, S2 and S3 Tables). The age and sex distribution of participants was representative of the resident population (S2 and S3 Tables).

The median village size at baseline was 108 (range 18 to 298) and median household size of 5 (interquartile range 4 to 7). At the 12-month visit, 418 (9.1%) baseline residents had

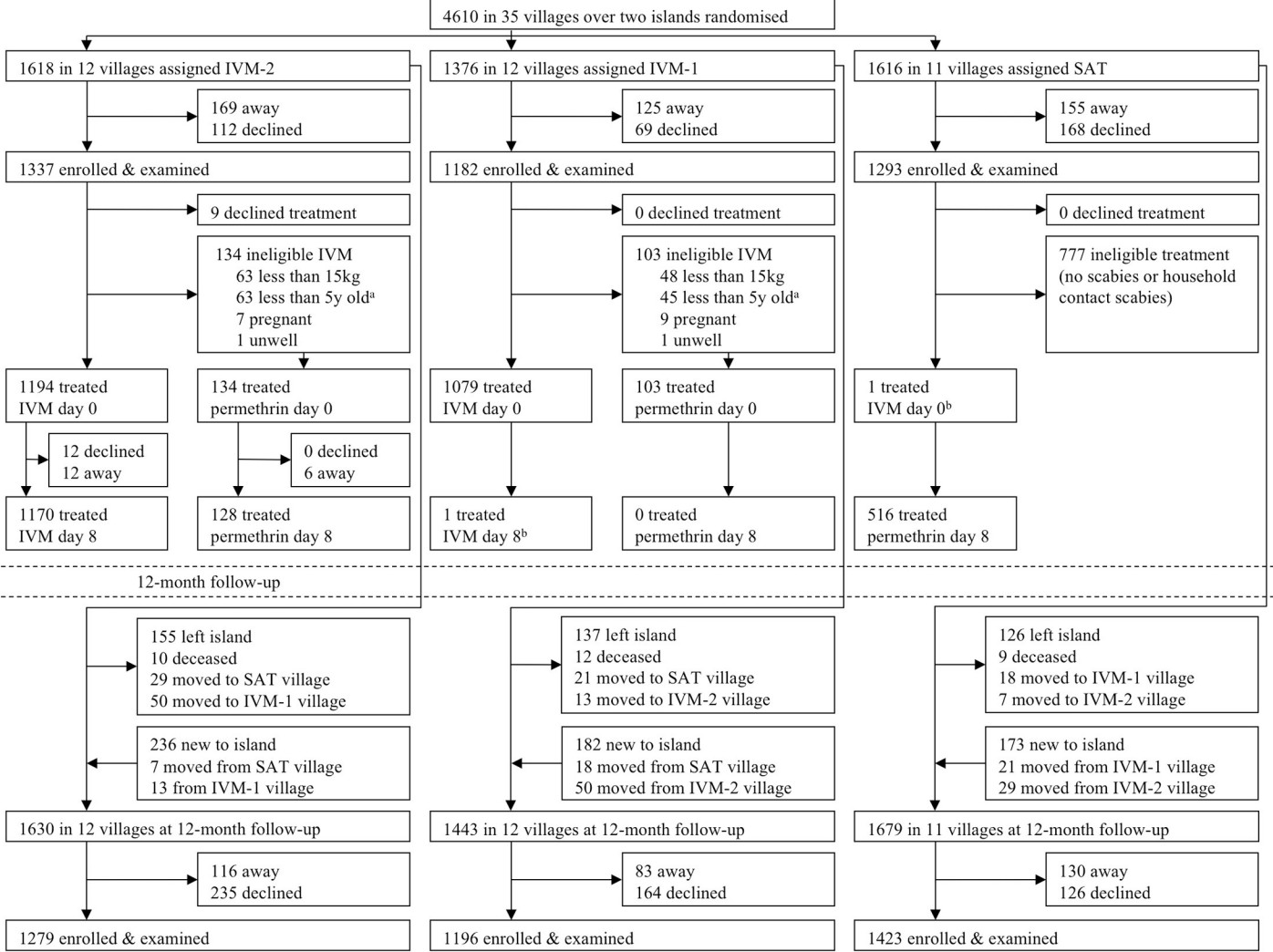

**Fig 2. Trial profile detailing village cluster randomisation, enrolment, and treatment at baseline and enrolment at 12-month follow-up.** IVM-1, one-dose ivermectin-based MDA; IVM-2, two-dose ivermectin-based MDA; MDA, mass drug administration; SAT, screen and treat with 1-dose permethrin to index cases of scabies and their household contacts. [a]Less than 5 years old and weight equal to or greater than 15 kilograms. [b]Treatment violation due to misclassification of participant's resident village.

**Table 1. Participant demographics at baseline and 12-month visits by treatment group.**

| | IVM-2 | | | | IVM-1 | | | | SAT | | | |
|---|---|---|---|---|---|---|---|---|---|---|---|---|
| | Baseline | | 12 months[a] | | Baseline | | 12 months[a] | | Baseline | | 12 months[a] | |
| | n | % | n | % | n | % | n | % | n | % | n | % |
| Population | 1,618 | | 1,630 | | 1,376 | | 1,443 | | 1,616 | | 1,679 | |
| Consented | 1,337 | 82.6 | 1,279 | 78.5 | 1,182 | 85.9 | 1,196 | 82.9 | 1,293 | 80.0 | 1,423 | 84.8 |
| **Sex** | | | | | | | | | | | | |
| Male | 694 | 51.9 | 693 | 54.2 | 599 | 50.7 | 617 | 51.6 | 673 | 52.0 | 751 | 52.8 |
| Female | 643 | 48.1 | 586 | 45.8 | 583 | 49.3 | 579 | 48.4 | 620 | 48.0 | 672 | 47.2 |
| **Age (years)** | | | | | | | | | | | | |
| Median (IQR) | 25 | (9–46) | 28 | (10–47) | 22 | (11–46) | 23 | (11–47) | 27 | (10–48) | 26 | (10–47) |
| <2 | 35 | 2.6 | 30 | 2.3 | 20 | 1.7 | 38 | 3.2 | 38 | 2.9 | 44 | 3.1 |
| 2–4 | 94 | 7.0 | 73 | 5.7 | 74 | 6.3 | 64 | 5.4 | 80 | 6.2 | 87 | 6.1 |
| 5–9 | 210 | 15.7 | 202 | 15.8 | 149 | 12.6 | 143 | 12.0 | 182 | 14.1 | 212 | 14.9 |
| 10–14 | 201 | 15.0 | 166 | 13.0 | 158 | 13.4 | 156 | 13.0 | 172 | 13.3 | 193 | 13.6 |
| 15–24 | 121 | 9.1 | 110 | 8.6 | 207 | 17.5 | 214 | 17.9 | 134 | 10.4 | 149 | 10.5 |
| 25–34 | 140 | 10.5 | 166 | 13.0 | 124 | 10.5 | 121 | 10.1 | 154 | 11.9 | 181 | 12.7 |
| 35–49 | 257 | 19.2 | 238 | 18.6 | 192 | 16.2 | 197 | 16.5 | 236 | 18.3 | 245 | 17.2 |
| 50–64 | 188 | 14.1 | 195 | 15.2 | 170 | 14.4 | 170 | 14.2 | 207 | 16.0 | 218 | 15.3 |
| ≥65 | 91 | 6.8 | 99 | 7.7 | 88 | 7.4 | 93 | 7.8 | 90 | 7.0 | 94 | 6.6 |

IQR, interquartile range; IVM-1, one-dose ivermectin-based MDA; IVM-2, two-dose ivermectin-based MDA; SAT, screen and treat with 1-dose permethrin to index cases of scabies and their household contacts.

[a]Participants allocated to treatment group of their current resident village in 2018.

permanently left the study sites, and 591 (12.4%) were newly arrived residents (Fig 2). A further 138 residents moved to a village in a different study group between visits. Of 3,884 participants surveyed at the 12-month visit, 2,089 (53.8%) reported having left the island at least once in the preceding 12 months.

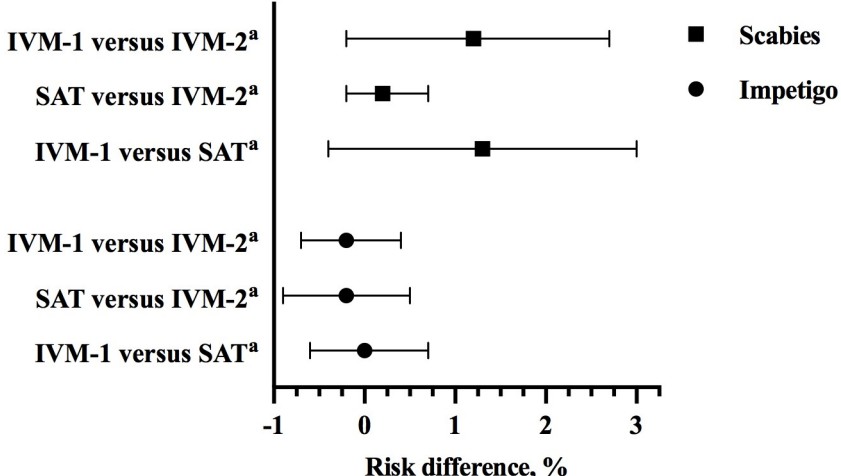

**Fig 3. Scabies prevalence risk difference and 95% CI between any 2 groups at 12 months.** CI, confidence interval; IVM-1, one-dose ivermectin-based MDA; IVM-2, two-dose ivermectin-based MDA; MDA, mass drug administration; SAT, screen and treat with 1-dose permethrin to index cases of scabies and their household contacts. Whiskers represent 95% CI around risk difference. [a]Reference treatment group.

Treatment coverage of the recorded resident population in the IVM-2 group was 82.1% for at least 1 dose and 80.2% for 2 doses. Two doses of ivermectin were given to 1,170 (87.5% of enrolled participants) and 2 doses of permethrin to 128 (9.6%) participants. In the IVM-1 group, resident population treatment coverage was 85.9%. In this group, ivermectin was given to 1,079 participants (91.3% of enrolled participants) and permethrin to 103 (8.7%; S2 Table). In the SAT group, there were 176 people found to have scabies with 340 uninfected household contacts, resulting in 516 participants receiving permethrin treatment (resident population treatment coverage of 31.9%, corresponding to 39.9% of enrolled participants; Fig 2).

Scabies was found in 513 individuals (13.5%) at baseline with similar prevalence between groups: IVM-2: 11.7%; IVM-1: 15.2%; SAT: 13.6% (Table 2). Scabies prevalence across villages ranged from 1.2% to 31.4% (S4 Table). Children aged less than 15 years had a higher prevalence compared to adults (S6 Table). There were no cases of crusted scabies.

At 12 months, the prevalence of scabies was substantially lower in all treatment groups: IVM-2: 1.3%; IVM-1: 2.7%; SAT: 1.1% (Table 2). The risk difference in scabies prevalence at 12 months for IVM-1 compared to IVM-2 group: 1.2%; for SAT compared to IVM-2: 0.2%; and IVM-1 compared to SAT: 1.3% (Fig 3). All villages except one (allocated to IVM-1) had a decrease in prevalence, and 16 out of 35 had no scabies cases detected at 12 months (S4 Table). Of the 64 cases detected at 12 months, 20 (31%) were in newly enrolled participants, 15 were in people who had scabies at baseline, and the remainder were in previously enrolled people without scabies at baseline.

At baseline, 82 (2.2%) participants had impetigo. The prevalence was similar across treatment groups (Table 2, S5 Table) and more prevalent in children aged less than 15 years (S7 Table). At 12 months, there was a decrease in impetigo prevalence in all groups to 1.0% or less. The risk difference in impetigo prevalence at 12 months for IVM-1 compared to IVM-2 group: −0.2%; for SAT compared to IVM-2: −0.2%; and IVM-1 compared to SAT: 0.0% (Fig 3). Of the 82 participants with impetigo at baseline, 62 (75.6%) had concurrent scabies, representing a population risk of impetigo attributable to scabies of 72.7% (95% CI 61.8 to 83.7). At 12 months, 9 of 27 cases of impetigo (33.3%) had scabies, representing a population risk of impetigo attributable to scabies of 28.6% (95% CI 7.7 to 49.4).

**Table 2. Scabies and impetigo prevalence at baseline and 12-month visits by treatment group.**

| | Prevalence at baseline[a] | | | | Prevalence at 12 months[a] | | | | Absolute reduction[b] | |
| --- | --- | --- | --- | --- | --- | --- | --- | --- | --- | --- |
| | N | n | % | (95% CI) | N | n | % | (95% CI) | % | (95% CI) |
| **IVM-2** | | | | | | | | | | |
| Scabies | 1,337 | 157 | 11.7 | (8.5–16.0) | 1,279 | 16 | 1.3 | (0.6–2.5) | 10.7 | (6.4–14.9) |
| Impetigo | 1,337 | 25 | 1.9 | (1.1–3.3) | 1,279 | 13 | 1.0 | (0.5–2.0) | 1.1 | (−0.4–2.5) |
| **IVM-1** | | | | | | | | | | |
| Scabies | 1,182 | 180 | 15.2 | (9.4–23.8) | 1,196 | 32 | 2.7 | (1.1–6.5) | 11.1 | (4.5–17.7) |
| Impetigo | 1,182 | 27 | 2.3 | (1.5–3.4) | 1,196 | 9 | 0.8 | (0.3–1.8) | 1.6 | (0.4–2.7) |
| **SAT** | | | | | | | | | | |
| Scabies | 1,293 | 176 | 13.6 | (7.9–22.4) | 1,423 | 16 | 1.1 | (0.6–2.0) | 10.1 | (4.7–15.4) |
| Impetigo | 1,293 | 30 | 2.3 | (1.3–4.1) | 1,423 | 5 | 0.4 | (0.2–0.7) | 2.0 | (0.7–3.2) |

CI, confidence interval; IVM-1, one-dose ivermectin-based MDA; IVM-2, two-dose ivermectin-based MDA; MDA, mass drug administration; SAT, screen and treat with 1-dose permethrin to index cases of scabies and their household contacts.

[a]Adjusted for clustering by village and stratification by island.

[b]Adjusted for clustering by village.

## Discussion

In our study of MDA for scabies in an endemic island setting, there was a significant reduction in the community prevalence of scabies and impetigo at 12 months in all 3 groups. One-dose ivermectin-based MDA and also a screen and treat approach using permethrin were not inferior to 2-dose ivermectin-based MDA.

The results from the MDA groups in our study are consistent with those of other studies in the Pacific in which ivermectin-based MDA led to substantially reduced prevalence of scabies and impetigo [5,8,19]. However, these previous studies used a 2-dose ivermectin-based MDA approach, and our study is the first randomised trial, to our knowledge, to demonstrate that community-wide treatment with a single dose is able to substantially reduce both burden and transmission, despite ivermectin having minimal ovicidal activity.

We also observed that screening for scabies and then providing 1 dose of permethrin treatment to individuals with clinical signs and their household contacts was highly effective in reducing community prevalence at 12 months. Direct comparison between our study and SHIFT is difficult because of higher baseline scabies prevalence in SHIFT study sites [5]. Nonetheless, the effectiveness of our screen and treat approach was higher than that reported for the corresponding standard care group in SHIFT, possibly because our study team dispensed permethrin, rather than referring to the local clinic for treatment as done in SHIFT [5,20].

While the effectiveness of all 3 groups was similar, we believe that a screen and treat approach would be impractical to implement as a large-scale public health strategy. This approach would be labour intensive and expensive, requiring a large workforce of highly skilled clinical examiners to screen all individuals within a population [20,21]. Furthermore, in high prevalence settings, identification and treatment of infected individuals and their household contacts may result in community treatment coverage approaching that of MDA.

There are a number of limitations to our study. First, diagnosis of scabies and impetigo was made by trained but nonexpert clinical examiners, without parasitological or microbiological confirmation, and these examiners were not blinded to treatment received. Nonexperts have been reported to be less sensitive in the diagnosis of mild cases of scabies compared to experts [21], and so it is possible that a proportion of cases of scabies may have been missed. Second, all participants in the study also received diethylcarbamazine and albendazole as MDA for lymphatic filariasis; however, neither of these medications have activity against the scabies mite or bacterial pathogens [13]. Third, while this study comparing MDA regimens was a nested trial, it was designed from the outset to be a stand-alone 3-arm, cluster randomised trial. Fourth, the baseline scabies prevalence in our study was lower than anticipated, likely due to geographical variation in prevalence, but this did not impact the statistical power of the trial.

This study was conducted on 2 small Pacific islands and may not be generalizable to larger populations and higher density settings. Another trial comparing 1 and 2 doses of ivermectin-based MDA for scabies is currently underway in the Solomon Islands [22], and before–after studies of the impact of filariasis MDA programmes that include a single dose of ivermectin on scabies are being conducted in a number of countries including Timor-Leste [23]. In addition, these results are limited to the first 12 months after MDA. Previous studies have demonstrated that 1 round of MDA may have a prolonged and sustained benefit, but we will not have data to determine if this will be replicated at our study sites [6,8].

Our study provides evidence from a randomised trial that 1-dose ivermectin-based MDA is noninferior to 2-dose and, therefore, may be adequate as a strategy for controlling scabies in endemic settings. More research is needed to support our finding, including in larger populations and in non-island settings. While we also found that the screen and treat strategy was equivalent, this approach would be impractical to implement at scale.

## Supporting information

**S1 Table. Medication dosing schedule by weight or age.**
(PDF)

**S2 Table. Baseline population and participant demographics and treatment coverage by village.** IQR, interquartile range; IVM-1, one-dose ivermectin-based MDA; IVM-2, two-dose ivermectin-based MDA; MDA, mass drug administration; SAT, screen and treat with 1-dose permethrin to index cases of scabies and their household contacts. [a]Villages 1–17 are on Rotuma; Villages 18–35 are on Gau; median village size 108. [b]Percentage of census population treated.
(PDF)

**S3 Table. Population and participant demographics at 12-month follow-up.** IQR: interquartile range; IVM-1, one-dose ivermectin-based MDA; IVM-2, two-dose ivermectin-based MDA; MDA, mass drug administration; SAT, screen and treat with 1-dose permethrin to index cases of scabies and their household contacts. [a]Villages 1–17 are on Rotuma; Villages 18–35 are on Gau; median village size 125.
(PDF)

**S4 Table. Scabies prevalence by village at baseline and 12 months.** CI, confidence interval; ICC, intracluster correlation; IVM-1, one-dose ivermectin-based MDA; IVM-2, two-dose ivermectin-based MDA; MDA, mass drug administration; SAT, screen and treat with 1-dose permethrin to index cases of scabies and their household contacts. [a]Villages 1–17 are on Rotuma; Villages 18–35 are on Gau. [b]One-sided 97.5% CI. [c]Adjusted for clustering by village and stratified by island. [d]Adjusted for clustering by village. The ICC coefficient for scabies at baseline was 0.120, and 12 months was 0.207.
(PDF)

**S5 Table. Impetigo prevalence by village at baseline and 12 months.** CI, confidence interval; ICC, intracluster correlation; IVM-1, one-dose ivermectin-based MDA; IVM-2, two-dose ivermectin-based MDA; MDA, mass drug administration; SAT, screen and treat with 1-dose permethrin to index cases of scabies and their household contacts. [a]Villages 1–17 are on Rotuma; Villages 18–35 are on Gau. [b]One-sided 97.5% CI. [c]Adjusted for clustering on village and stratified by island. [d]Adjusted for clustering on village. The ICC coefficient for impetigo at baseline was 0.07, and 12 months was 0.06.
(PDF)

**S6 Table. Scabies prevalence at baseline and 12 months by treatment and demographic groups.** IQR: interquartile range; IVM-1, one-dose ivermectin-based MDA; IVM-2, two-dose ivermectin-based MDA; MDA, mass drug administration; SAT, screen and treat with 1-dose permethrin to index cases of scabies and their household contacts. [a]Participants allocated to treatment group of their current resident village in 2018.
(PDF)

**S7 Table. Impetigo prevalence at baseline and 12 months by treatment and demographic groups.** IQR: interquartile range; IVM-1, one-dose ivermectin-based MDA; IVM-2, two-dose ivermectin-based MDA; MDA, mass drug administration; SAT, screen and treat with 1-dose permethrin to index cases of scabies and their household contacts. [a]Participants allocated to treatment group of their current resident village in 2018.
(PDF)

**S1 Fig. Randomisation and treatment flowchart.** D, day; IVM, ivermectin; MDA, mass drug administration. [a]Ineligible for treatment if not found to have scabies and no household contacts had scabies.
(PDF)

**S1 Data. Anonymized individual level data at baseline and 12 months.**
(XLSX)

**S1 Protocol. Community-based safety of 2-drug (diethylcarbamazine and albendazole) versus 3-drug (ivermectin, diethylcarbamazine, and albendazole) therapy for lymphatic filariasis in Fiji—Protocol v6.0 6 August 2019.**
(PDF)

**S1 Methods. Sample size calculation.**
(PDF)

**S1 CONSORT checklist. CONSORT 2010 checklist of information to include when reporting a cluster randomised trial.**
(PDF)

## Acknowledgments

Our sincere thanks to the communities on Rotuma and Gau islands, Fiji. We acknowledge the support of the Fiji Ministry of Health and Medical Services (FMOHMS), the Fiji Ministry of iTaukei Affairs, the Fiji Ministry of Education, Heritage and Arts, and the Rotuman Council. The following people made significant contributions to the study: Humphrey Biutilomaloma and Uraia Kanito, Fiji Data Managers, Murdoch Children's Research Institute (MCRI); Aminiasi Koroivueti and Sarah Gwonyoma, Fiji Project Officers, MCRI; Patrick Lammie and Andrew Majewski, The Taskforce for Global Health; Joshua Bogus and Rachel Anderson, Global Project Managers, Death to Onchocerciasis and Lymphatic Filariasis (DOLF), St. Louis; Kobie O'Brian, Global Data Manager, DOLF, St. Louis; Catherine Bjerum, Laboratory and Good Clinical Practice Trainer, Case Western Reserve University; and the rest of the Fiji Integrated Therapy study team.

## Author Contributions

**Conceptualization:** Myra Hardy, Josaia Samuela, Mike Kama, Lucia Romani, Margot J. Whitfeld, Christopher L. King, Gary J. Weil, Daniel Engelman, Leanne J. Robinson, John M. Kaldor, Andrew C. Steer.

**Data curation:** Myra Hardy.

**Formal analysis:** Myra Hardy, Anneke C. Grobler, Andrew C. Steer.

**Funding acquisition:** Gary J. Weil, Andrew C. Steer.

**Investigation:** Myra Hardy.

**Methodology:** Myra Hardy, Tibor Schuster, Daniel Engelman, John M. Kaldor, Andrew C. Steer.

**Project administration:** Myra Hardy, Josaia Samuela, Mike Kama, Meciusela Tuicakau, Lucia Romani, Andrew C. Steer.

**Resources:** Josaia Samuela, Mike Kama, Meciusela Tuicakau, Lucia Romani, Christopher L. King, Gary J. Weil.

**Software:** Myra Hardy, Lucia Romani.

**Supervision:** Josaia Samuela, Mike Kama, Meciusela Tuicakau, Margot J. Whitfeld, Christopher L. King, Gary J. Weil, Leanne J. Robinson, John M. Kaldor, Andrew C. Steer.

**Validation:** Myra Hardy, Tibor Schuster, Anneke C. Grobler.

**Visualization:** Myra Hardy, Andrew C. Steer.

**Writing – original draft:** Myra Hardy.

**Writing – review & editing:** Myra Hardy, Josaia Samuela, Mike Kama, Meciusela Tuicakau, Lucia Romani, Margot J. Whitfeld, Christopher L. King, Gary J. Weil, Tibor Schuster, Anneke C. Grobler, Daniel Engelman, Leanne J. Robinson, John M. Kaldor, Andrew C. Steer.

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
