## [Editor Report · Decision Letter 0]

16 Jun 2021

Dear Dr Hardy, 

Thank you for submitting your manuscript entitled "Community control strategies for scabies: a cluster randomised non-inferiority trial" for consideration by PLOS Medicine.

Your manuscript has now been evaluated by the PLOS Medicine editorial staff and I am writing to let you know that we would like to send your submission out for external peer review.

Please re-submit your manuscript within two working days, i.e. by Jun 18 2021 11:59PM.

Kind regards,

Beryne Odeny

Associate Editor

PLOS Medicine

---

## [Decision Letter · Decision Letter 1]

13 Aug 2021

Dear Dr. Hardy,

Thank you very much for submitting your manuscript "Community control strategies for scabies: a cluster randomised non-inferiority trial" (PMEDICINE-D-21-02600R1) for consideration at PLOS Medicine. 

Your paper was evaluated by an academic editor with relevant expertise and sent to independent reviewers, including a statistical reviewer. The reviews are appended at the bottom of this email and any accompanying reviewer attachments can be seen via the link below:

[LINK]

In light of these reviews, we will not be able to accept the manuscript for publication in the journal in its current form, but we would like to invite you to submit a revised version that addresses the reviewers' and editors' comments fully. You will appreciate that we cannot make a decision about publication until we have seen the revised manuscript and your response, and we expect to seek re-review by one or more of the reviewers. 

We hope to receive your revised manuscript by Sep 03 2021 11:59PM. Please email us (plosmedicine@plos.org) if you have any questions or concerns.

Please let me know if you have any questions, and we look forward to receiving your revised manuscript. 

Sincerely,

Richard Turner PhD, for Beryne Odeny, 

Senior editor, PLOS Medicine

rturner@plos.org

You mention in the article metadata that all data are included in the ms and supplementary files - are you able to include anonymized patient-level date in excel files or similar?

PLOS Medicine style is to use "sex" rather than "gender", where appropriate.

Please quote aggregate demographic details for participants in the abstract; along with some key details of villages/clusters.

In the abstract and throughout the paper, please quote p values alongside 95% CI, where available.

Please combine the "Methods" and "Findings" subsections of your abstract. 

The final sentence of the new combined subsection should begin "Study limitations include ..." or similar, and should quote 2-3 of the study's main limitations. 

Please remove the information on funding from the abstract. In the event of publication, this information will appear in the article metadata, via entries in the submission form. 

Please restructure the "Author summary" so that each of the three subsections consists of around 3 bulleted points, each consisting of no more than 1 or 2 short sentences. 

Please remove the "Role of the funding source" section from the Methods. 

Please avoid claims such as "the first", in the final paragraph for example, and where needed add "to our knowledge" or similar.

Throughout the text, please locate reference call-outs prior to punctuation and remove spaces from within the square brackets (e.g., "... over 26,000 [7,8].").

Please confirm that the map in fig. 1 can be published under a CC BY licence, in the event of publication. 

Comments from the reviewers:

*** Reviewer #1: 

Community control strategies for scabies: a cluster randomised non-inferiority trial

This is an important report on a cluster randomised trial for prevention and control of scabies. The authors do make a case for the scientific gap is whether a one-dose ivermectin-based mass drug administration (MDA) would not be inferior to a two-dose ivermectin-based MDA. So the authors designed an open-label cluster RCT with 3 arms:

- (IVM-2) Ivermectin-based treatment plus DEC and albendazole followed by a second dose of ivermectin 8 days later

- (IVM-1) Ivermectin-base treatment plus DEC and albendazole

- (SAT) Clinical screen for scabies and treatment (offered as well DEC and albendazole) 

General comments:

1. The primary outcome in this trial is absolute reduction of prevalence from baseline to 12 months. OK to use the IVM-2 as the reference group.

a) The sample size calculation was done under the assumption of higher baseline prevalence and larger absolute reductions. This deserves note and more discussion. In fact the results section almost obscures this because it focuses more on the secondary outcome see below.

b) Please add more notes on how the simulation for sample size calculation was done. 

c) Please add information on how cluster correction/adjustment was done (on table 2). The lines 213 to 218 do not explain this.

2. For the secondary outcome [scabies/impetigo prevalence at 12 months]. The authors state that they used "risk differences". However, the lines 222 and 223 say that they use a GLM regression model with binomial family and log-link. This model is also known as log-binomial in the literature and is typically used to obtain relative risks (or proportion ratios), a clear relative measure of association. I would suggest the authors change the terminology from "risk differences" to "prevalence ratio" (or "proportion ratio" etc...)

Please see more comments on the CONSORT checklist. 

CONSORT checklist

1) Outcomes (Item 6a)

Completely defined pre-specified primary outcome measure including how and when it was assessed

R: Yes it is clearly defined. Lines 203 and 206. Absolute reduction in community prevalence of scabies between baseline and 12 months. This is in percentage.

2) Sample size (Item 7a)

How sample size was determined

R: Between lines 207 and 210 there is an explanation of how the sample size was computed. Expected effectiveness is collected from the literature review. A simulation procedure of 1000 confidence intervals with Bonferroni correction was used to determine the sample size of 24 clusters. However, I cannot find further details of this approach in the attached protocol. They assumed a power of 80% (β = 20%) and α error level of 5%. Another key assumption was that the scabies prevalence would be between 25 to 35% (their results show that overall baseline prevalence ranged between 11.7 to 15.2%, few communities/clusters reached above 25%). 

3) Sequence generation (Item 8a)

Method used to generate random allocation sequence

Does the description make it clear if the "assigned intervention is determined by a chance process and cannot be predicted"?

R: Yes. Lines 171 to 173. An independent statistician generated the allocation using Stata. 

4) Allocation concealment (Item 9)

Mechanism used to implement random allocation sequence (such as sequentially numbered containers), describing any steps taken to conceal the sequence until interventions were assigned

Is it clear how the care provider enrolling participants was made ignorant of the next assignment in the sequence (different from blinding)? Possible methods can rely on centralised or "third-party" assignment (i.e., use of a central telephone randomisation system, automated assignment system, sealed containers).

R: The authors state there was allocation concealment in line 173. 

5) Blinding (Item 11a)

If done, who was blinded after assignment to interventions (for example, participants, care providers, those assessing outcomes)

Is it clear if (1) healthcare providers, (2) patients, and (3) outcome assessors are blinded to the intervention? General terms such as "double-blind" without further specifications should be avoided.

R: There was no blind. This is said in line 174.

6) Outcomes and estimation (Item 17a/b)

For the primary outcome, results for each group, and the estimated effect size and its precision (such as 95% confidence intervals)

Is the estimated effect size and its precision (such as standard deviation or 95% confidence intervals) for each treatment arm reported? When the primary outcome is binary, both the relative effect (risk ratio, relative risk) or odds ratio) and the absolute effect (risk difference) should be reported with confidence intervals.

R: For the primary outcome (absolute reduction of scabies or impetigo prevalence from baseline to 12 months follow up, reported on table 2) no measure of association is presented. The reporting text is a bit unclear on this but it is implicit (according to how the sample size was estimated) that the intention is to check the overlapping of the 95% confidence interval. Per arm baseline and 12 month prevalence are presented and differences corrected for clustering. However, is not described in the manuscript how this adjustment/correction per clustering was done for this particular analysis (that lead to table 2). 

7) Harms (Items 19)

All important harms or unintended effects in each group

Is the number of affected persons in each group, the severity grade (if relevant) and the absolute risk (e.g. frequency of incidence) reported? Are the number of serious, life threatening events and deaths reported? If no adverse event occurred this should be clearly stated.

R: This was not reported and I do not think this is relevant for this cluster RCT.

8) Registration (Item 23)

Registration number and name of trial registry

Is the registry and the registration number reported? If the trial was not registered, it should be explained why.

R: The trial was registered prospectively lines 51 and 52. 

Clinitrials.gov NCT03177993 and ANZCTR N12617000738325

9) Protocol (Item 24)

Where trial protocol can be accessed

Is it stated where the trial protocol can be assessed (e.g. published, supplementary file, repository, directly from author, confidential and therefore not available)?

R: The protocol is added as supplementary material.

10) Funding (Item 25)

Sources of funding and other support (such as supply of drugs) and role of funders

Are (1) the funding sources, and (2) the role of the funder(s) described?

R: There is a section declaring the funding sources (lines 66 to 73, only in the abstract). And there is a section for the role of funding source (lines 229 to 232). 

*** Reviewer #2: 

Scabies is an important problem and this will be an excellent addition to the literature on its control. The study design is simple and appropriate. It appears that the work was sound. The paper is concise and clear.

I have no major comments and only a few very minor ones for the authors' consideration.

Line 153: Having "residing in 35 villages" at the end of this sentence seems odd. Suggest add to the front of the sentence, which would then read, "The trial was conducted in 2017 and 2018 in 35 villages…"

Line 237, Table 1 and elsewhere: suggest replace "sex" with "gender", which is more likely to be what was recorded by field teams.

Line 257: I am not sure how to reconcile these two sentences, which at first reading seem to be inconsistent: "Treatment coverage in the IVM-2 group was 82.1% for at least one dose and 80.2% for two doses. Two doses of ivermectin were given to 1170 (87.5% enrolled) and two doses of permethrin to 128 (9.6%) participants." Are the authors using the resident population as the denominator in the first sentence and consenting participants in the second sentence?

Line 267: suggest add "aged" in front of "less" in the sentence, "Children less than 15 years…"

Line 295: "Of the 82 participants with impetigo at baseline, 62 (75.6%) had concurrent scabies, representing a population risk of impetigo attributable to scabies of 72.7% (95% CI 61.8-83.7, p<0.0001). At 12 months, 9 of 27 cases of impetigo (33.3%) had scabies, representing a population risk of impetigo attributable to scabies of 28.6% (95% CI 7.7-49.4, p=0.007)." Suggest just use two significant digits here, since in my view that's all that's justifiable given the data.

Line 320: "This approach would be labour-intensive and expensive, requiring a large workforce of highly skilled clinical examiners to screen all individuals within a population." I agree with this statement. It would be made more powerfully if the authors could add either here or in the methods at line 178 some more detail of how the examiners in the study were trained and assessed for competency.

Line 324: "approaching that of an MDA strategy": suggest simplify to "approaching that of MDA".

*** Reviewer #3: 

Review comments

PMEDICINE-D-21-02600R1 

Community control strategies for scabies: a cluster randomised non-inferiority trial

General comments

Scabies is a neglected tropical disease that is endemic in many parts of the world. The WHO Road Map for NTDs 2030 has set targets for the control/ eradication of scabies and this calls for renewed efforts and approaches to control. Optimisation of MDA approaches for scabies treatment is therefore an essential tool to aid the attainments of the targets set out in the road map. The authors are therefore to be commended for undertaking this important study.

The authors performed a cluster randomised, non-inferiority, open-label, three group unblinded study to compare the effectiveness of 3 control strategies on community prevalence of scabies on 2 Fijian islands 12 months after the intervention. The rationale for the study is clearly set out in the manuscript. The authors are to be commended for undertaking this important work which enables us answer some key questions on scabies control.

The manuscript is very well written.

The methodology is clear and results are very well presented. the findings and their public health implications have been well discussed

The manuscript has been reported in conformance with the CONSORT checklist. The main concern is regarding adverse events. No account is given regarding adverse events (if any). This is especially important as the trial was nested with co-administration with DEC and albendazole for LF. It is important to make some comment on the safety of this co-administered regimen

CONSORT checklist

1) Outcomes (Item 6a)

Completely defined pre-specified primary outcome measure including how and when it was assessed

Is it clear (1) what the primary outcome is (usually the one used in the sample size calculation), (2) how it was measured (if relevant; e.g. which score used), (3) at what time point, and (4) what the analysis metric was (e.g. change from baseline, final value)?

Well done

2) Sample size (Item 7a)

How sample size was determined

Is there a clear description of how the sample size was determined, including (1) the estimated outcomes in each group; (2) the α (type I) error level; (3) the statistical power (or the β (type II) error level); and (4) for continuous outcomes, the standard deviation of the measurements? Yes

3) Sequence generation (Item 8a)

Method used to generate random allocation sequence

Does the description make it clear if the "assigned intervention is determined by a chance process and cannot be predicted"? Yes

4) Allocation concealment (Item 9)

Mechanism used to implement random allocation sequence (such as sequentially numbered containers), describing any steps taken to conceal the sequence until interventions were assigned

Is it clear how the care provider enrolling participants was made ignorant of the next assignment in the sequence (different from blinding)? Possible methods can rely on centralised or "third-party" assignment (i.e., use of a central telephone randomisation system, automated assignment system, sealed containers).

5) Blinding (Item 11a)

If done, who was blinded after assignment to interventions (for example, participants, care providers, those assessing outcomes)

Is it clear if (1) healthcare providers, (2) patients, and (3) outcome assessors are blinded to the intervention? General terms such as "double-blind" without further specifications should be avoided. N/A

6) Outcomes and estimation (Item 17a/b)

For the primary outcome, results for each group, and the estimated effect size and its precision (such as 95% confidence intervals)

Is the estimated effect size and its precision (such as standard deviation or 95% confidence intervals) for each treatment arm reported? When the primary outcome is binary, both the relative effect (risk ratio, relative risk) or odds ratio) and the absolute effect (risk difference) should be reported with confidence intervals.

Done 

7) Harms (Items 19)

All important harms or unintended effects in each group

Is the number of affected persons in each group, the severity grade (if relevant) and the absolute risk (e.g. frequency of incidence) reported? Are the number of serious, life threatening events and deaths reported? If no adverse event occurred this should be clearly stated.

Not included in current report. Authors should include some information on adverse events (if any) and how they were handled

8) Registration (Item 23)

Registration number and name of trial registry

Is the registry and the registration number reported? If the trial was not registered, it should be explained why. 

Yes; trial was registered with Clinitrials.gov NCT03177993 and ANZCTR N12617000738325).

9) Protocol (Item 24)

Where trial protocol can be accessed

Is it stated where the trial protocol can be assessed (e.g. published, supplementary file, repository, directly from author, confidential and therefore not available)?

Yes; attached as a supplementary file

10) Funding (Item 25)

Sources of funding and other support (such as supply of drugs) and role of funders

Are (1) the funding sources, and (2) the role of the funder(s) described? Yes

*** Reviewer #4: 

Scabies review

Accept 

This is a well planned, executed, analysed and presented study. 

However, I believe it would be worthwhile if the authors could make some points regarding the following issue:

1. Is there any evidence that the nurses who conducted the clinical examinations were able to accurately diagnose scabies? This can sometimes be difficult, even for dermatologists. Are the authors able to provide more information about their training?

Although not directly related to the study aims and findings, from the perspective of scabies control in general I think it would be helpful if the authors could (in the discussion)

1. Briefly mention that follow up incidence studies would be useful; for example in 2023 and 2028. This could yield information regarding the longevity of the effect. With this knowledge, an optimal frequency of mass treatments could be estimated.

2. Briefly comment on the very low incidence rates in adults greater than ~35 years of age. Does this age group need to be mass treated? If cost is an issue, then halving the number of doses administered may permit mass treatments to be done more often: for example, every 5 years instead of every decade. (And it is certainly significant that the authors have been able to show that they can reduce costs by half by simply avoiding the second dose).

*** Reviewer #5:

Statistical review

This paper reports a cluster randomised trial comparing different strategies for mass drug administration to reduce prevalence of scabies. The trial shows that the intervention arms were non-inferior to the control arm. The trial is generally reported well and the paper written clearly. 

I have some comments, which I have provided below.

1. Abstract, Line 60 and results: I would recommend providing a p-value for non-inferiority in addition to the 95% CI.

2. Page 11 - how was stratification by island incorporated into the randomisation? Using stratified blocks? From this paragraph I would presume that villages agreed to participate prior to knowing the allocation and none dropped out after allocation - if this is not the case then please make this clear.

3. Page 13 - for the sample size calculation it appears the power was averaging over the different prevalence scenarios. I don't think enough detail on this is provided to allow reproduction of the sample size. Also, could the authors add what the assumed clustering coefficient (e.g. ICC) was? Lastly, Bonferroni correction was mentioned, but I did not see how this linked with using the 95% CI for non-inferiority (which implies no correction, unless the overall error rate is 5% one-sided).

4. Page 13-14: for the statistical analysis, two approaches are outlined - using the difference between follow-up and baseline prevalence, and then using a generalised linear approach. 

For the former approach, could more details be provided on the model assumed. Presumably it is using summary data per cluster rather than individual patient data? 

For the latter approach, is the baseline prevalence data used at all?

5. I would imagine it's normal for this type of study, but the differential consent rates per arm are noticeable in table 1. It would be useful if the authors could discuss implications of this on how robust the results are. I suspect they are robust as it would require a huge difference between non-consenting and consenting participants to change the conclusions.

6. Line 281 - many of the new cases were in participants newly in the study at 12 months - does this mean they were unlikely to have actually benefited from the interventions?

7. Line 297 - it's not clear to me what the significant p-value here means. Is this that it's significantly higher than 0? Can the estimated PAR and CI ever be negative?

8. There are several secondary outcome measures mentioned on the clinicaltrials.gov registration page that are not reported in this paper. I would recommend that these are mentioned in the outcomes section and either reported here or a statement that they are to be reported separately provided.

9. The protocol provided appears to be for the larger study rather than the scabies substudy specifically. If there is no specific protocol for the substudy, then pointing out the relevant sections of the protocol in the Supplementary text description may be useful.

James Wason

***

[LINK]

---

## [Decision Letter · Decision Letter 2]

7 Oct 2021

Dear Dr. Hardy,

Thank you very much for re-submitting your manuscript "Community control strategies for scabies: a cluster randomised non-inferiority trial" (PMEDICINE-D-21-02600R2) for review by PLOS Medicine.

I have discussed the paper with my colleagues and the academic editor and it was also seen again by two reviewers. I am pleased to say that provided the remaining editorial and production issues are dealt with we are planning to accept the paper for publication in the journal.

[LINK]

We look forward to receiving the revised manuscript by Oct 14 2021 11:59PM.   

Sincerely,

Beryne Odeny, 

PLOS Medicine

plosmedicine.org

Requests from Editors:

1) Abstract:

a) Please structure your abstract using the PLOS Medicine headings (Background, Methods and Findings, Conclusions). The second subheading should be “Methods and Findings”

b) Please include p-values alongside 95% CIs

c) Please replace the subtitle “Interpretation” with “Conclusions”

Comments from Reviewers:

Reviewer #3: The authors have addressed all queries that were raised.

Reviewer #5: Thank you to the authors for addressing my previous comments well. I have no further issues to raise.

[LINK]

---

## [Editor Report · Decision Letter 3]

14 Oct 2021

Dear Dr Hardy, 

On behalf of my colleagues and the Academic Editor, Dr. Lorenz von Seidlein, I am pleased to inform you that we have agreed to publish your manuscript "Community control strategies for scabies: a cluster randomised non-inferiority trial" (PMEDICINE-D-21-02600R3) in PLOS Medicine.

PRESS

Sincerely, 

Beryne Odeny 

PLOS Medicine